# Electrocardiographic interpretation by emergency medical services professionals in Saudi Arabia: A cross sectional study

**Mohammed Abdullah Alalwan**[1], **Talal Alshammari**[1], **Hassan Alawjan**[1], **Hassan Alkhayat**[1], **Ahmed Alsaleh**[1], **Ibrahim Alamri**[1], **Alaa Aldubaikel**[2], **Jaber Alqahtani**[3], **Ahmad Alrawashdeh**[4], **Saeed Alqahtani**[5]*

1 Department of Emergency Medical Care, College of Applied Medical Sciences, Imam Abdulrahman Bin Faisal University, Dammam, Saudi Arabia, 2 Department of Academic Affairs and Training, Saudi Red Crescent Authority, Dammam, Saudi Arabia, 3 Department of Respiratory Care, Prince Sultan Military College for Health Sciences, Dhahran, Saudi Arabia, 4 Department of Allied Medical Sciences, Jordan University of Science and Technology, Irbid, Jordan, 5 Department of Emergency Medical Services, Prince Sultan Military College for Health Sciences, Dhahran, Saudi Arabia

* saeeddewairem@gmail.com

**Data Availability Statement:** The data available in Harvard Dataverse: https://doi.org/10.7910/DVN/IVB1QG.

## Abstract

### Background

Management of acute myocardial infarction (AMI) and cardiac arrhythmias in prehospital settings is largely determined by providers of emergency medical services (EMS) who can proficiently interpret the electrocardiography (ECG). The aim of this study was to assess the ECG competency of EMS providers in Saudi Arabia.

### Methods

Between Aug and Sep 2022, we invited all EMS providers working for the Saudi Red Crescent Authority in Makkah, Riyadh, and Sharqiyah regions to complete a cross-sectional survey. The survey was used to assess the ability of EMS providers to interpret 12 ECG strips. Characteristics and ECG competency were summarized using descriptive statistics. Differences in ECG competency across paramedics with lower and higher qualifications were assessed.

### Results

During the study period, 231 participants completed the survey, and all were included. The overall mean age was 33.4, and most participants were male (94.8%). Nearly half of the participants were paramedics with an associate degree and 46.4% were paramedics with higher degrees. The average rate of correct answers to the 12 ECG strips was 43.3% (95% CI: 35.4%, 51.3%). Atrial flutter, ventricular fibrillation, atrial fibrillation, 3rd degree heart block, and ventricular tachycardia were identified by 52.8%, 60.2%, 42.0%, 40.7%, and 49.4% of the participants, respectively. The strip with an AMI was identified by 41.1%, while a pathological Q wave and ventricular extrasystole were identified by 19.1% and 24.7%, respectively. Paramedics with higher qualifications were as 28.0%-61.0% more likely to

**Funding:** The author(s) received no specific funding for this work.

**Competing interests:** The authors have declared that no competing interests exist.

correctly interpret the 12 ECG strips compared to those with an associate degree (p-value across all variables was $\leq 0.001$).

## Conclusion

While the majority of participants in our region were unable to correctly answer the 12 ECG questionnaire, paramedics with higher qualifications were. Our study indicates that there is a need for evidenced-based ECG curricula targeting different levels of EMS professionals.

## Background

Ischemic heart disease (IHD) accounted for 9.4 million deaths worldwide in 2021 [1]. Early screening and prevention interventions for individuals at high risk of IHD in primary health-care settings do not reduce mortality [2]. While many people normally live with and cope with heart disease, the disease can progress to life-threatening conditions [3]. Acute myocardial infarction (AMI) and cardiac arrythmias are examples of these and are often encountered by emergency medical services (EMS) personnel in prehospital settings. Using the 12-lead electro-cardiography (ECG), EMS personnel can improve the survival and quality of life outcomes of patients following AMI and cardiac arrhythmias [4, 5]. However, not every prehospital care provider is competent in interpreting the ECG [6–9].

A number of studies have been carried out in the US, Canada, and Australia to assess the competency of EMS personnel in diagnosing patients with AMI using the 12-lead ECG [10]. In Boston, US, paramedics with basic and advanced life support skills were able to identify 80% of patients with confirmed AMI [8]. Similarly, primary care paramedics were able to accurately diagnose patients with AMI in 79% of the cases in Simcoe County, Canada [7]. Additionally, mobile intensive care paramedics from Ambulance Victoria, Australia, diagnosed 58% of AMI cases [9]. Although prehospital care providers from these regions have not correctly detected 20%-40% of the cases, such estimates were derived from studies published before 2013. They were also pooled from studies conducted across regions with developed EMS systems. As such, it is unclear whether paramedics in a developing EMS system such as the EMS of the Saudi Red Crescent Authority (SRCA) would reach better or similar figures. In addition, since a large proportion of prehospital care providers in the SRCA are either para-medics with associate or higher degrees, it is not known if the level of education would have an impact on the ECG interpretation skills.

This study aimed to assess the ability of prehospital care providers in the SRCA to correctly interpret ECG strips using a previously published ECG competency survey [11]. Differences in ECG competency across levels of paramedic education will also be examined.

## Methods

### Study design

This was a cross-sectional study. All EMS personnel working for the SRCA in Makkah, Riyadh, and Sharqiyah regions were invited through their SRCA email account to voluntarily complete an electronic survey between 27th August 2022 and 21st September 2022. In Saudi Arabia, much of the resources of EMS (53.8%) are allocated to those regions, servicing more than 21,700,000 people (67.6%) of the Saudi population (n = 32,175,200 in 2022). Each EMS provider included in this study has formally consented to participate at the start of the survey. Information for each participant was obtained and maintained in a de-identified data format.

Participants from other regions were excluded. This study has ethics approval from the Institutional Review Board of SRCA (No: 22-66E).

Since the number of our population is a continuous variable, we decided to use Adam's 2020 formula for the sample size calculation [12]. Across the three regions, there were ≤1500 EMS personnel ($N$). At a value of 0.03 for the degree of accuracy ($e$), 4.0 for the number of standard deviations ($\rho$), 1.96 for the t-value with a two-tailed 95% confidence interval ($t$), and 0.0612 for the adjusted margin of error ($\epsilon$), our sample size ($n$) should include ≈ 227 participants. Where $\epsilon = \left(\frac{\rho(e)}{t}\right)$ $and$ $n = N/(1 + N\epsilon^2)$.

## Settings

The SRCA is the primary national provider of EMS in Saudi Arabia. Prehospital care ambulance units manned with basic and advanced life support personnel respond to emergency cases. Personnel in EMS operate under the clinical practice guidelines of the SRCA (www.srca.org.sa). All patients with chest pain or epigastric discomfort or presenting with signs and symptoms of acute coronary syndrome must be assessed before leaving the scene using the 12-lead ECG. The ECG strips are then transmitted through an electronic communication platform to the online medical director for further medical consultations.

Responding to cases of cardiovascular disease and acute coronary syndrome by EMS personnel is within the National EMS Scope of Practice (www.srca.org.sa). All EMS personnel working for the SRCA are registered with the Saudi Health Council for Health Specialties (SHCFHS) [13]. An EMS provider must pass a written test or an oral interview in his or her related field to be registered with the SHCFHS. There are some training courses provided by the SRCA to all EMS clinicians to improve the quality of prehospital care provided. This includes basic and advanced life support and ECG training courses.

## Survey

The 12-item questionnaire developed by Coll-Badell et al. in 2017 was administered to our study population with minor modifications. Permission to use the survey was obtained from the primary author. The survey was originally used to assess the ability of emergency nurses to interpret the 12 ECG strips. The survey is comprised of two sections. Section one was designed to collect demographical data such as age, sex, location, years of experience, and previous ECG training courses. Section two was designed to assess the ECG knowledge and interpretation skills of the participants. In this section, participants were asked to answer two broad questions about the waveform shape of the ECG and interpret 10 ECG strips with different levels of complexity. One correct answer out of four possible answers was developed for each question.

## Statistical analysis

The characteristics and ECG knowledge and interpretation skills of the study population were summarized using descriptive statistics. Continuous variables were reported as means and standard deviations. Categorical variables were reported as counts and percentages. We stratified our population into two groups to assess differences in ECG interpretation skills across these stratifications. Paramedics with an associate degree were compared to paramedics with bachelor's and master's degrees using the Chi-square test ($X^2$). Since previous ECG training courses and ECG modes of instruction may produce some confounding effects on our assessment, we performed a sensitivity analysis. We assessed whether such variables would significantly differ between the two groups using the $X^2$ test. A two-tailed p-value of $< 0.05$ was

considered statistically significant. All statistical analyses were carried out using STATA statistical software, version 16.0 (Statacorp, College Station, Texas, USA).

## Results

During the study period, 231 EMS personnel completed the survey, and all were included in the final analyses.

### Baseline characteristics

Table 1 presents the baseline characteristics of the study population. The overall mean age was 33.4, and most participants were male (94.8%). The majority of respondents were from the Riyadh region (60.6%), and 34.6% had 1–5 years of work experience. Paramedics with an associate degree represented nearly half of the study population (48.1%), while paramedics with undergraduate and graduate degrees represented the other half (46.4%). About 60.0% of the participants reported that they had undertaken an ECG training course. Of those, 44.9% reported that the ECG training course they had taken was ≤ 1 year ago, and 68.4% reported that the ECG mode of instruction was face-to-face. The majority of the participants received less than 10 hours of ECG training (69.8%).

### Interpretation of the ECG strips

The ability of the EMS providers to interpret the ECG is presented in Table 2. Approximately sixty percent of the participants were able to list the correct order of the waveform shape on the ECG, and 52.4% were able to assess the value of the P-wave presentation on the strip. Atrial flutter, ventricular fibrillation, atrial fibrillation, 3$^{rd}$ degree heart block, and ventricular tachycardia were identified by 52.8%, 60.2%, 42.0%, 40.7%, and 49.4% of the study population, respectively. Forty-one percent of the participants were able to identify the strip with acute myocardial infarction, and < 25.0% were able to identify strips with a pathological Q wave and ventricular extrasystole. On average, the rate of participants who were able to correctly identify the ECG strips was 43.3% (95% CI: 35.4%, 51.3%). The indicated figures in Table 1 are shown in Fig 1.

### Differences in ECG interpretation between the EMS groups

Table 3 compared personnel in EMS with an associate degree to those with bachelor's and master's degrees. Across the 12-item questionnaire, paramedics with higher qualifications were as 28.0%-61.0% more likely to correctly answer the questions and interpret the ECG strips compared to those with an associate degree (p-value across all variables was ≤ 0.001). Differences in previous ECG training courses and ECG modes of instruction between the two groups were not statistically significant (Table 4).

## Discussion

In this cross-sectional study, the average rate of correct answers to the 12 ECG strips for all participants (n = 231) was 43.3% (95% CI: 35.4%, 51.3%). Of the 12 ECG questionnaire, more than 74.0% of the study population were not able to correctly identify cardiac rhythms with a pathological Q wave and ventricular extrasystole. Paramedics with bachelor's and master's degrees (n = 107), on average, had a rate of correct answers of 65.4% (95% CI: 54.1%, 76.8%) compared to 25.2% (95% CI: 18.3%, 32.1%) for paramedics with an associate degree (n = 111).

Our participants had an overall lower rate of correct answers (43.3%) compared to the rate reported in a study with a similar study design. The authors of the original survey tool we used

**Table 1. Descriptive statistics of the study population.**

| Variables | Statistics, n = 231 |
|---|---|
| Age, mean (SD) | 33.4 (6.4) |
| Sex, n (%) | |
| Male | 219 (94.8) |
| Female | 12 (5.2) |
| Region, n (%) | |
| Riyadh | 117 (60.6) |
| Makkah | 56 (29.0) |
| Sharqiyah | 20 (10.4) |
| Years of experience in prehospital care, n (%) | |
| < 1 year | 19 (8.2) |
| 1–5 years | 80 (34.6) |
| 6–10 years | 60 (26.0) |
| 11–20 years | 51 (22.1) |
| >20 years | 21 (9.1) |
| EMS provider, n (%) | |
| Paramedic with an associate degree | 111 (48.1) |
| Paramedic with a bachelor's degree | 96 (41.6) |
| Paramedic with a master's degree | 11 (4.8) |
| Medical doctor | 1 (0.4) |
| Nurse | 8 (3.5) |
| Other | 4 (1.7) |
| Undertook ECG training course | |
| No | 95 (41.1) |
| Yes | 136 (58.9) |
| Last ECG training course taken, n (%) | |
| ≤ 1 year | 61 (44.9) |
| 2-5years | 60 (44.1) |
| >5 years | 15 (11.0) |
| Mode of instruction, n (%) | |
| Online | 34 (25.0) |
| Face to face | 93 (68.4) |
| Partial face to face | 9 (6.6) |
| Duration of the course, n (%) | |
| <10 hours | 95 (69.8) |
| 10–20 hours | 33 (24.3) |
| >20 hours | 8 (5.9) |

**Abbreviation**. SD: Denotes Standard Deviation; EMS: Denotes Emergency Medical Services; ECG: Denotes Electrocardiography.

in our study have reported a rate of correct answers of 86.8% [11]. In that study, all participants were undergraduate nurses working in emergency departments. It is common for many healthcare professionals with undergraduate degrees to receive some formal education in cardiology and ECG interpretation. It is also possible for nurses to have gained more experience in ECG interpretation through their rich exposure to cardiac cases, which are often seen in emergency rooms or hospital settings. However, our finding seems comparable to a study from Sweden. In that study, 54.0% of ambulance nurses successfully provided the correct

**Table 2. Interpretation of the ECG strips by the study population.**

| No | Question | Correct answer | n = 231 (%) |
|---|---|---|---|
| 1 | What is the correct order of the ECG? | It is P, QRS, T, PR, ST, and U. | 136 (58.9) |
| 2 | What do you think if the P wave does not appear on the ECG strip? | I think there is a conduction problem between atriums. | 121 (52.4) |
| 3 | What do you think the rhythm in Fig 1a might be? | It might be an atrial flutter. | 122 (52.8) |
| 4 | How would you react to the rhythm in Fig 1b? | I will ask for help without leaving the case. It is likely ventricular fibrillation. | 139 (60.2) |
| 5 | What is your interpretation of the rhythm in Fig 1c? | It is likely atrial fibrillation. | 97 (42.0) |
| 6 | What is your interpretation of the rhythm in Fig 1d if you know that this strip belongs to a patient with precordial pain for > eight hours? | It is likely a pathological Q wave. | 44 (19.1) |
| 7 | What pathology does the rhythm in Fig 1e present? | The rhythm in Fig 1e likely presents a third-degree heart block. | 94 (40.7) |
| 8 | What does the rhythm in Fig 1f present, given that the patient has a history of surgery as a result of acute myocardial infarction? | The rhythm in Fig 1f likely presents a ventricular tachycardia. | 114 (49.4) |
| 9 | What is your interpretation of the rhythm in Fig 1g if you know that this strip belongs to a 52-year-old male with moderate precordial pain two hours ago? The patient has a history of hypertension and type II diabetes. | It is likely an acute myocardial infarction. | 95 (41.1) |
| 10 | What is your interpretation of the rhythm in Fig 1h, which belongs to a young athletic adult who experiences chest pain after completing his exercise three hours earlier? | It is likely a normal sinus rhythm. | 86 (37.2) |
| 11 | What is your interpretation of the rhythm in Fig 1i? This rhythm belongs to a patient with digitalis intoxication. | It is likely a ventricular extrasystole. | 57 (24.7) |
| 12 | What is your interpretation of the rhythm in Fig 1j? This rhythm belongs to a 30-year-old female with palpitation, chest discomfort, and shortness of breath. | It is likely an atrial tachycardia. | 96 (41.6) |

answers to the 16 ECG strips they had been provided with [14]. Although, most of our study population were paramedics with different levels of education, our participants and participants from the Swedish study received no formal ECG training right before taking the test and were not often exposed to cardiac cases.

Prehospital care providers in our region were more likely to misinterpret the ECG rhythms with premature ventricular contraction and patterns indicating a wide range of differential diagnoses. Our finding is similar to that reported in other regions of Saudi Arabia. Across 427 medical students from Umm Al-Qura University, Saudi Arabia, the ventricular extrasystole and pathological Q wave were not identified by nearly 75.1% of the participants [15]. Medical interns were also not able to correctly diagnose patients with such rhythms (75.7%) in a study carried out across 14 medical universities in Saudi Arabia [16]. Conversely, our finding differs from that reported in the Iran region. In that report, 55.7% were not able to identify ECG strips with a pathological Q wave and ventricular extrasystole [17]. However, 74.0% of the included population were nurses and physicians. Hospital-based working healthcare professionals are likely to be more trained and experienced in ECG interpretation compared to those who are not [18]. While this could partly explain the deficiency of ECG interpretation across our population, there are also some concerns that the ECG training curriculum is often not standardized and tailored to a specific profession with a range of mastery levels [18, 19]. It is possible that our participants were not up to the advanced levels of ECG interpretation.

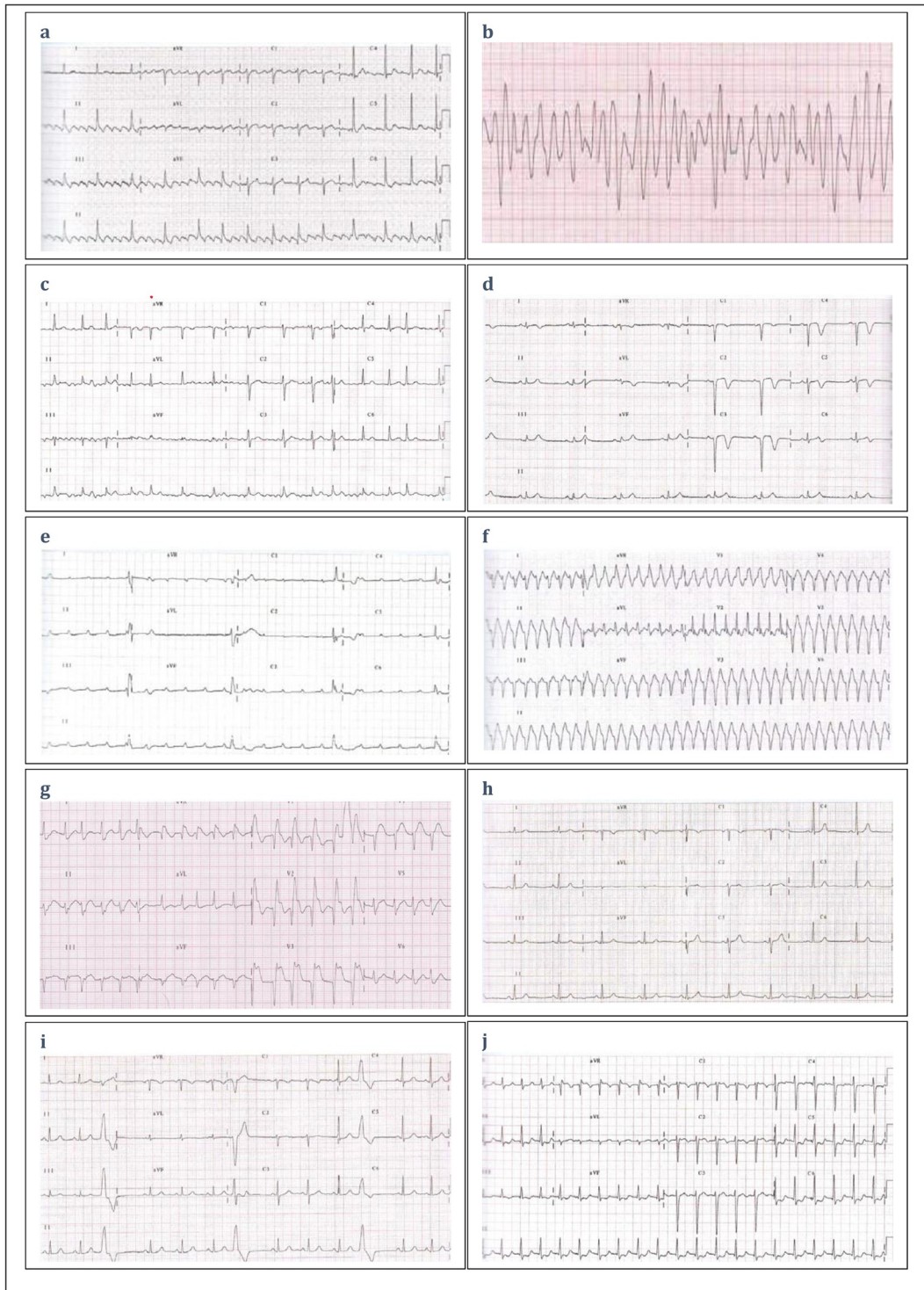

**Fig 1. The electrocardiographic strips.**

**Table 3. Differences in ECG interpretation between the two groups.**

| No | Correct answers | EMS | | P-value |
|---|---|---|---|---|
| | | With an associate degree, n = 111 (%) | With bachelor's and master's degrees, n = 107 (%) | |
| 1 | It is P, QRS, T, PR, ST, and U. | 50 (45.1) | 85 (79.4) | < 0.001 |
| 2 | I think there is a conduction problem between atriums. | 35 (31.5) | 84 (78.5) | < 0.001 |
| 3 | It might be an atrial flutter. | 30 (27.0) | 91 (85.1) | < 0.001 |
| 4 | I will ask for help without leaving the case. It is likely ventricular fibrillation. | 43 (38.8) | 94 (87.9) | < 0.001 |
| 5 | It is likely atrial fibrillation. | 19 (17.1) | 75 (70.1) | < 0.001 |
| 6 | It is likely a pathological Q wave. | 12 (10.8) | 30 (28.0) | = 0.001 |
| 7 | The rhythm in Fig 1e likely presents a third-degree heart block. | 21 (18.9) | 70 (65.4) | < 0.001 |
| 8 | The rhythm in Fig 1f likely presents a ventricular tachycardia. | 37 (33.3) | 75 (70.1) | < 0.001 |
| 9 | It is likely an acute myocardial infarction. | 32 (28.8) | 62 (57.9) | < 0.001 |
| 10 | It is likely a normal sinus rhythm. | 17 (15.3) | 68 (63.6) | < 0.001 |
| 11 | It is likely a ventricular extrasystole. | 13 (11.7) | 41 (38.3) | < 0.001 |
| 12 | It is likely an atrial tachycardia. | 27 (24.3) | 65 (60.8) | < 0.001 |

Paramedics with higher levels of education were 40.2% more proficient in interpreting the ECG than those with lower levels. This is often common across many healthcare disciplines. For example, paramedic students in year four from King Saud University, Saudi Arabia, were 23.0% more competent in ECG interpretation than those in year two [20]. In medicine, the accuracy of ECG interpretation by cardiologists was 32.9%, 19.1%, and 6.4% greater than that of medical students, residents, and practicing physicians, respectively [18]. In the nursing field, the level of knowledge in ECG interpretation was 27.3% higher in nurses with undergraduate degrees relative to nurses with diplomas [21]. Our finding suggests that higher education is an important factor in paramedics' performance when dealing with cardiac patients. In a previous report from Riyadh, Saudi Arabia, the 12-lead ECG strip with inferior MI was correctly identified by 67.1% of advanced life support paramedics [6]. Undergraduate and graduate education for paramedics is the norm in many regions, such as the UK, Australia, and New-Zealand [22]. There is also some evidence from the US indicating that paramedic education is shifting from associate degrees to bachelor's and master's degrees between 2017 and 2019 [23].

Our study has some important implications and recommendations. The ECG is a simple diagnostic test used to record cardiac rhythms on a screen or paper graph [24]. Many

**Table 4. Differences in ECG training and mode of instruction between the two groups.**

| Variables | EMS | | P-value |
|---|---|---|---|
| | With an associate degree, n = 59 | With bachelor's and master's degrees, n = 73 | |
| Last ECG training course taken, n (%) | | | 0.082 |
| ≤ 1 year | 20 (33.9) | 37 (50.7) | |
| > 1 year | 39 (66.1) | 36 (49.3) | |
| Mode of instruction, n (%) | | | 0.526 |
| Online | 39 (66.1) | 53 (72.6) | |
| Face-to-face | 15 (25.4) | 17 (23.3) | |
| Partial face-to-face | 5 (8.5) | 3 (4.1) | |

healthcare professionals, such as medical doctors, nurses, and paramedics, use this test to detect a wide range of abnormal cardiac rhythms or conditions [19]. As a result, they are required to be well educated, trained, and experienced in ECG interpretation [19]. Importantly, they are also required to make sense of the clinical presentations associated with such abnormalities, necessitating the need to perform the ECG test [19]. After this, they can make meaningful decisions. However, the decision to intervene varies from one profession to another and within one profession.

For example, pathological Q waves can only be managed by cardiologists [25]. Additionally, the prehospital AMI can only be managed by intensive care paramedics or highly trained EMS personnel with the aid of online medical direction [26]. Despite this, differences in qualifications and levels of training among healthcare professionals are not considered in ECG curricula and training courses [19]. To date, many healthcare providers are saturated with too much information in the ECG that is beyond their scope of practice. Interpreting the ECG without linking the ECG findings to patients' care is challenging for any healthcare professional. It is more challenging for those who are required to make rapid clinical decisions, such as EMS personnel.

The primary role of EMS is to provide basic, advanced, and critical care life support in prehospital settings. The personnel of many EMS systems are paramedics with basic, advanced, and critical care qualifications and training skills. Since there is no ECG competency standard for the EMS populations, it is plausible to create one through a Delphi study. Findings from this study can inform the development of ECG curricula and training courses specific to each EMS level. After proper training on ECG interpretation, follow-up studies across levels of EMS can be carried out to determine the time needed for a refresher ECG training course".

## Limitations

This study has some potential limitations. First, our data were obtained from regions that are highly populated and have a high demand for EMS. Some EMS stations across those regions receive more calls compared to other stations. It is therefore possible that some of our participants were less or more exposed to cardiac cases and arrhythmias. Differences in exposure to such cases may have produced some confounding effect that could underestimate or overestimate our rates. Second, prehospital care providers in our region generally receive formal education in cardiology and are trained to interpret the ECG, but the same level of teaching and training for all participants cannot be confirmed. Our participants may have a low, medium, or high level of literacy in ECG or a mix of all. As such, although our finding indicates that paramedic with higher degrees were better than those with an associate degree, we are not completely certain. Third, while ongoing medical training is offered by the SRCA to all EMS personnel, some rural EMS personnel in a previous study reported that they lack enough access to ongoing medical training and education [27]. Unfortunately, rurality was not considered as part of the analyses to explain some of our findings. Fourth, the survey was emailed to participants so they could independently complete it. It is therefore difficult to conclude that no participant received any assistance in completing the survey. Fifth, participation in our study was voluntary. It is therefore possible that those who opted to participate were more competent in ECG interpretation than those who did not. A limitation that could overestimate our findings.

## Conclusions

Prehospital care providers in our region were 43.3% competent in ECG interpretation. The ECG strips with a premature ventricular contraction and pathological Q wave were likely to be

misdiagnosed by the majority of our study population. Paramedics with higher qualifications were 40.2% more proficient in ECG interpretation relative to those with a lower qualification. There is a need for evidenced-based ECG curricula targeting different levels of EMS professionals. Such curricula should be undertaken on a regular basis, and this regularity should also be evidence based.

## Supporting information

**S1 Checklist. STROBE statement—Checklist of items that should be included in reports of observational studies.**
(DOCX)

## Acknowledgments

We are very thankful to the SRCA officials and staff members from the research unit of the SRCA for their unlimited support of our study. Special thanks go to Ms. Randa Amadhari and Dr. Yousef Alsofayan for reviewing the final version of our manuscript.

## Author Contributions

**Conceptualization:** Mohammed Abdullah Alalwan, Talal Alshammari.

**Data curation:** Mohammed Abdullah Alalwan, Ahmad Alrawashdeh, Saeed Alqahtani.

**Formal analysis:** Mohammed Abdullah Alalwan, Ahmad Alrawashdeh, Saeed Alqahtani.

**Investigation:** Mohammed Abdullah Alalwan, Hassan Alawjan, Hassan Alkhayat, Ahmed Alsaleh, Ibrahim Alamri, Saeed Alqahtani.

**Methodology:** Mohammed Abdullah Alalwan, Talal Alshammari, Ahmed Alsaleh, Jaber Alqahtani, Ahmad Alrawashdeh.

**Project administration:** Alaa Aldubaikel.

**Supervision:** Talal Alshammari, Alaa Aldubaikel, Jaber Alqahtani, Saeed Alqahtani.

**Visualization:** Talal Alshammari, Alaa Aldubaikel, Jaber Alqahtani, Ahmad Alrawashdeh, Saeed Alqahtani.

**Writing – original draft:** Mohammed Abdullah Alalwan, Saeed Alqahtani.

**Writing – review & editing:** Talal Alshammari, Ahmed Alsaleh, Alaa Aldubaikel, Jaber Alqahtani, Ahmad Alrawashdeh, Saeed Alqahtani.

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
