## [Decision Letter · Decision Letter 0]

8 Aug 2023

PONE-D-23-21375Electrocardiographic interpretation by emergency medical services professionals in Saudi Arabia: A cross sectional studyPLOS ONE

Dear Dr. Alalwan,

Thank you for submitting your manuscript to PLOS ONE. After careful consideration, we feel that it has merit but does not fully meet PLOS ONE’s publication criteria as it currently stands. Therefore, we invite you to submit a revised version of the manuscript that addresses the points raised during the review process.

We look forward to receiving your revised manuscript.

Kind regards,

Ibrahim Marai, MD

Academic Editor

PLOS ONE

Journal Requirements:

Reviewers' comments:

Reviewer's Responses to Questions

**Comments to the Author**

1. Is the manuscript technically sound, and do the data support the conclusions?

Reviewer #1: Yes

Reviewer #2: Yes

2. Has the statistical analysis been performed appropriately and rigorously? 

Reviewer #1: I Don't Know

Reviewer #2: Yes

3. Have the authors made all data underlying the findings in their manuscript fully available?

Reviewer #1: Yes

Reviewer #2: Yes

4. Is the manuscript presented in an intelligible fashion and written in standard English?

Reviewer #1: Yes

Reviewer #2: No

5. Review Comments to the Author

Reviewer #1: This is a cross sectional study aimed to assess competency in ECG interpretation among EMS providers in three regions in Saudi Arabi. The research is well done, and the manuscript is clearly presented with appropriate methodology, results, discussion, and conclusions. However, I have a minor comment regarding previous studies. I believe the study of Alrumayh et al., 2022 ,which also assessed paramedic ability in recognizing 12-lead Electrocardiogram (ECG) with ST-segment Elevation myocardial infarction (STEMI) in Saudi Arabia, can be used in the discussion section to compare the results of the two studies.

Reviewer #2: This paper reports on the findings of a study examining the competency of EMS providers in Saudi Arabia in assessing an ECG. While it could be perceived that the focus on Saudi Arabia is narrow, I believe that the results are contextualised against findings from other countries, and are likely to be of wider interest and therefore relevant to this journal.

In general, the manuscript is easy to understand and largely free of typographical errors (although there are some), but it would benefit from a thorough edit to remove grammatical inconsistencies. There are a number of incomplete sentences, odd punctuation and occasional incorrect word choices that all require remediation.

I do have some other suggestions for the authors to consider, and these are detailed below:

- Is there a reason why those three regions (Makkah, Riyadh and Sharqiyah) were chosen? Are they representative of all regions of Saudi Arabia? I recognise that you note this as a potential limitation at the end of the manuscript, so it would be good to include a justification as to why you made this decision initially.

- There are noted differences in the literature regarding both outcomes and training between urban and rural EMS in Saudi Arabia - do these three regions have a mix of both urban and rural services? Is this likely to have any ramifications in terms of your recommendations?

- Following on from this point regarding geographic location, some literature has shown a greater proportion of EMS staff with associate (versus advanced) degrees in rural areas of Saudi Arabia. Was rurality considered as part of the analysis, and/or as an explanation for some findings?

- I note that the final sample included 1 medical doctor, 8 nurses and 4 'others'. I wonder whether there is really any value in including them when their numbers are so small and any findings from this group would therefore be irrelevant? I am also interested whether there was a considerable overlap between these 13 and the 12 female participants?

- Do you think that there is any bias in the final participant sample in terms of those who chose to participate versus those who did not?

- Based on your findings, do you have any specific suggestions for what a standardised ECG curriculum should look like, and how often EMS staff should do refreshers?

6. PLOS authors have the option to publish the peer review history of their article (what does this mean?). If published, this will include your full peer review and any attached files.

Reviewer #1: **Yes: **Mohammed S Aljohani

Reviewer #2: No

---

## [Author Response · Author response to Decision Letter 0]

15 Sep 2023

21 Aug 2023

Saeed Alqahtani

PhD, MSc, & BSc in Paramedicine & MSc in Medicine (Clinical Epidemiology)

Prince Sultan Military College of Health Sciences & Military Medical Services

P.O. Box: 946 Al Dharan 31932

Dr. Ibrahim Marai

Academic Editor

PLOS ONE

Dear Dr. Ibrahim,

RE: Manuscript PONE-D-23-21375 with the title “Electrocardiographic interpretation by emergency medical services professionals in Saudi Arabia: A cross sectional study”

Thank you for the opportunity to have our manuscript considered by PLOS ONE. On behalf of the investigators, I wish to thank the editor and reviewers for their valued contributions and provide the following responses to their appraisal:

a. Adhering to journal requirements:

We amended the format of the title page, abstract, and body as per PLOS ONE style requirements. We also restructured Tables 1, 2, 3, and 4 in the manuscript to meet PLOS ONE guidelines. For Table 2, we removed all embedded figures. These figures were then grouped into one figure (Fig. 1) and inserted below Table 2 with appropriate indications. 

We amended our data availability statement to “No – some restrictions will apply”. We also stated that “the dataset analysed during the current study is available from the corresponding author on a reasonable request”.

3. PLOS requires an ORCID ID for the corresponding author in Editorial Manager on papers submitted after December 6th, 2016. Please ensure that you have an ORCID ID and that it is validated in Editorial Manager.

We added the ORCID ID for the corresponding author in the Editorial Manager. The ORCID ID for the corresponding author is 0000-0003-4077-6455.

4. Please review your reference list to ensure that it is complete and correct. If you have cited papers that have been retracted, please include the rationale for doing so in the manuscript text or remove these references and replace them with relevant current references. Any changes to the reference list should be mentioned in the rebuttal letter that accompanies your revised manuscript. If you need to cite a retracted article, indicate the article’s retracted status in the References list and also include a citation and full reference for the retraction notice.

The reference list has been reviewed and updated according to PLOS ONE guidelines.

b. Response to comments provided by reviewer 1:

1. This is a cross sectional study aimed to assess competency in ECG interpretation among EMS providers in three regions in Saudi Arabi. The research is well done, and the manuscript is clearly presented with appropriate methodology, results, discussion, and conclusions.

We thank the reviewer for the positive appraisal of our work.

2. However, I have a minor comment regarding previous studies. I believe the study of Alrumayh et al., 2022 ,which also assessed paramedic ability in recognizing 12-lead Electrocardiogram (ECG) with ST-segment Elevation myocardial infarction (STEMI) in Saudi Arabia, can be used in the discussion section to compare the results of the two studies.

We thank the reviewer for this important comment. Comparing our results with those reported by Alrumah et al. (2022) was difficult for two reasons.1 First, paramedics with associate degrees were excluded. Second, we could not calculate the overall rate of correct answers because the denominator was unstable. Across all rhythms, there were a number of missing responses. However, in the last paragraph of our discussion section, we have used some information from the suggested study to support our finding, which is related to the importance of paramedic higher education.

 “Our finding suggests that higher education is an important factor for paramedics’ performance when dealing with cardiac patients. In a previous report from Riyadh, Saudi Arabia, the 12-lead ECG strip with inferior MI was correctly identified by 67.1% of the advanced life support paramedics.1 Undergraduate and graduate education for paramedics is the norm in many regions such as the UK, Australia, and New-Zealand”. 

c. Response to comments provided by reviewer 2:

1. This paper reports on the findings of a study examining the competency of EMS providers in Saudi Arabia in assessing an ECG. While it could be perceived that the focus on Saudi Arabia is narrow, I believe that the results are contextualised against findings from other countries and are likely to be of wider interest and therefore relevant to this journal.

We thank the reviewer for the positive appraisal of our work.

2. In general, the manuscript is easy to understand and largely free of typographical errors (although there are some), but it would benefit from a thorough edit to remove grammatical inconsistencies. There are a number of incomplete sentences, odd punctuation, and occasional incorrect word choices that all require remediation.

We thank the reviewer for this important comment. We have made a thorough edit throughout the manuscript to remove grammatical inconsistencies, make sure sentences are complete, and use correct word choices. 

3. I do have some other suggestions for the authors to consider, and these are detailed below:

I. Is there a reason why those three regions (Makkah, Riyadh and Sharqiyah) were chosen? Are they representative of all regions of Saudi Arabia? I recognise that you note this as a potential limitation at the end of the manuscript, so it would be good to include a justification as to why you made this decision initially.

We thank the reviewer for asking this question. We chose those regions because we believe that they represent the majority of the Saudi regions. According to the 2019 statistics from the Saudi Red Crescent Authority (SRCA), 53.8% of the EMS workforce is located in those regions (See Table 1). In addition, according to the 2022 population statistics, 67.6% of the Saudi population resides in those regions (See Table 2). 

Table 1. Workforce of EMS in Saudi Arabia (2019).

No Region Ambulances, n % Ambulances in the first three regions and remaining 10 regions, n (%)

1 Makkah 310 22.5 742 (53.8%)

2 Riyadh 259 18.8 

3 Sharqiyah 173 12.5 

4 Madinah 101 7.3 637 (46.2%)

5 Assir 107 7.8 

6 Qassim 84 6.1 

7 Jazan 52 3.8 

8 Tabuk 53 3.8 

9 Hail 51 3.7 

10 Najran 50 3.6 

11 Aljouf 45 3.3 

12 Albaha 49 3.6 

13 Southern Boarder 45 3.3 

Total 1,379 100 

Access to this information can be found at https://www.srca.org.sa/en/statistics/annual-report/.

Table 2. Population statistics in Saudi Arabia (2022).

No Region Population, n % Population in the first three regions and remaining 10 regions, n (%)

1 Riyadh 8,591,748 26.7 21,738,465 (67.6%)

2 Makkah 8,021,463 24.9 

3 Sharqiyah 5,125,254 15.9 

4 Madinah 2,137,983 6.6 10,436,759 (32.4%)

5 Aseer 2,024,285 6.3 

6 Jazan 1,404,997 4.4 

7 Qassim 1,336,179 4.2 

8 Tabuk 886,036 2.8 

9 Hail 746,406 2.3 

10 Aljouf 595,822 1.9 

11 Najran 592,300 1.8 

12 Southern Boarder 373,577 1.2 

13 Albaha 339,174 1.1 

Total 32,175,224 100 

Access to this information can be found at https://database.stats.gov.sa/home/indicator/535. 

In our study design section, after the second sentence, we added the following sentence:

 “In Saudi Arabia, much of the resources of EMS (53.8%) are allocated to those regions, servicing more than 21,700,000 people (67.6%) of the Saudi population (n=32,175,200 in 2022)”. 

II. There are noted differences in the literature regarding both outcomes and training between urban and rural EMS in Saudi Arabia - do these three regions have a mix of both urban and rural services? Is this likely to have any ramifications in terms of your recommendations?

We thank the reviewer for this important comment. In fact, there are reported differences regarding both outcomes and training between urban and rural EMS in Saudi Arabia. The length of stay in the hospital and intensive care unit is longer for patients who were treated by rural EMS than those who were treated by Urban EMS.2 Additionally, patients treated by rural EMS received fewer interventions compared to those treated by urban EMS.2 Differences in the level of education and scope of practice between rural and urban EMS could partly explain some of the observed differences. In rural regions, the majority of EMS personnel are presumably paramedics with associate degrees. Some rural EMS personnel from the Riyadh region reported that they lack ongoing medical training.3 

However, ongoing medical training and education is offered to all EMS personnel by the SRCA in the form of face-to-face or online training (Aldubaikel, A. (2023) E-mail to Saeed Alqahtani, 6 July 2023).3 In addition, permission to practice as a healthcare provider in Saudi Arabia is often limited to a certain number of years (usually two to four years).4 Once this permission expires, the healthcare provider must provide evidence of ongoing medical training and education in his or her related field and be recertified in some courses, such as the basic and advanced life support courses.4 Self-development is primarily the responsibility of the healthcare provider.

The answer to whether these three regions have a mix of both urban and rural services is yes. The answer to whether this is likely to have any ramifications in terms of our recommendations is no. While it is reasonable to recommend an ECG training program for remote EMS personnel, we are not sure whether competency in ECG interpretation differs between rural and urban EMS. Unfortunately, rurality was not a variable of interest when we designed our study. 

III. Following on from this point regarding geographic location, some literature has shown a greater proportion of EMS staff with associate (versus advanced) degrees in rural areas of Saudi Arabia. Was rurality considered as part of the analysis, and/or as an explanation for some findings?

We thank the reviewer for this important comment. As described above, rurality was not a variable of interest when we designed our study. We have added rurality as a potential limitation in our limitations section. 

“Third, while ongoing medical training is offered by the SRCA to all EMS personnel, some rural EMS personnel in a previous study reported that they lack enough access to ongoing medical training and education.3 Unfortunately, rurality was not considered as part of the analyses to explain some of our findings”. 

IV. I note that the final sample included 1 medical doctor, 8 nurses and 4 'others'. I wonder whether there is really any value in including them when their numbers are so small and any findings from this group would therefore be irrelevant? I am also interested whether there was a considerable overlap between these 13 and the 12 female participants?

We thank the reviewer for this important comment. We agree with the reviewer that including non-paramedic professionals is of less significance to our study because their numbers were small. We could also have excluded non-paramedic professionals from participating in our study. However, the EMS of the SRCA includes a number of healthcare professions, such as nurses and medical doctors, in addition to paramedics.4 Additionally, the SRCA has introduced new professions in 2021 to provide prehospital care, such as EMS-physicians and first medical responders.5 (Pages 9, 13-14 At the time of conducting the survey, we were not sure about the profession distribution of the included population. At the final analyses, we displayed the results as we found them. In Table 3, we presented the correct answers to the 12-item questionnaire for the medical doctor, nurses, and others. 

Table 3. Interpretation of the ECG strips by the medical doctor, nurses, and others.

No Correct answers Doctors, n=1 Nurses, n=8 Others, n=4

01 It is P, QRS, T, PR, ST, and U. 1 (100) 0 (0.0) 0 (0.0)

02 Conduction problem between atriums. 1 (100) 1 (12.5) 0 (0.0)

03 Atrial flutter. 1 (100) 0 (0.0) 0 (0.0)

04 Ventricular fibrillation. 1 (100) 1 (12.5) 0 (0.0)

05 Atrial fibrillation. 1 (100) 2 (25.0) 0 (0.0)

06 Pathological Q wave. 1 (100) 1 (12.5) 0 (0.0)

07 Third degree heart block. 1 (100) 2 (25.0) 0 (0.0)

08 Ventricular tachycardia. 1 (100) 1 (12.5) 0 (0.0)

09 Acute myocardial infarction. 0 (0.0) 1 (12.5) 0 (0.0)

10 Normal sinus rhythm. 0 (0.0) 1 (12.5) 0 (0.0)

11 Ventricular extrasystole 1 (100) 2 (25.0) 0 (0.0)

12 Atrial tachycardia. 1 (100) 3 (37.5) 0 (0.0)

Our response to the second part of this comment is that there was no considerable overlap between the 13 non-paramedic professionals and the 12 female participants. In Table 4, 91.7% (n = 11) of the female participants were paramedics with a bachelor’s degree.

Table 4. Female distribution in relation to EMS professions

Variable Female, n=12 Male, n=219

EMS provider, n (%) 

Paramedic with an associate degree 0.0 111 (50.7)

Paramedic with a bachelor’s degree 11 (91.7) 85 (38.8)

Paramedic with a master’s degree 0.0 11 (5.0)

Medical doctor 1.0 (8.3) 0.0

Nurse 0.0 8 (3.7)

Other 0.0 4 (1.8)

V. Do you think that there is any bias in the final participant sample in terms of those who chose to participate versus those who did not?

We thank the reviewer for this important comment. Yes, there is a potential bias. We added the decision to participate as a potential limitation in our limitations section.

“Fifth, participation in our study was voluntary. It is therefore possible that those who opted to participate were more competent in ECG interpretation than those who did not. A limitation that could overestimate our findings”.

VI. Based on your findings, do you have any specific suggestions for what a standardised ECG curriculum should look like, and how often EMS staff should do refreshers?

We thank the reviewer for this important comment. Yes, we have. We added the following paragraphs at the end of our discussion section to describe the problems with the current ECG curriculum and provide suggestions to resolve them.

“Our study has some important implications and recommendations The ECG is a simple diagnostic test used to record cardiac rhythms on a screen or paper graph.6 Many healthcare professionals, such as medical doctors, nurses, and paramedics, use this test to detect a wide range of abnormal cardiac rhythms or conditions.7 As a result, they are required to be well educated, trained, and experienced in ECG interpretation.7 Importantly, they are also required to make sense of the clinical presentations associated with such abnormalities, necessitating the need to perform the ECG test.7 After this, they can make meaningful decisions. However, the decision to intervene varies from one profession to another and within one profession. 

For example, pathological Q waves can only be managed by cardiologists.8 Additionally, the prehospital AMI can only be managed by intensive care paramedics or highly trained EMS personnel with the aid of online medical direction.9 Despite this, differences in qualifications and levels of training among healthcare professionals are not considered in ECG curricula and training courses.7 To date, many healthcare providers are saturated with too much information in the ECG that is beyond their scope of practice. Interpreting the ECG without linking the ECG findings to patients’ care is challenging for any healthcare professional. It is more challenging for those who are required to make rapid clinical decisions, such as EMS personnel. 

The primary role of EMS is to provide basic, advanced, and critical care life support in prehospital settings. The personnel of many EMS systems are paramedics with basic, advanced, and critical care qualifications and training skills. Since there is no ECG competency standard for the EMS populations, it is plausible to create one through a Delphi study. Findings from this study can inform the development of ECG curricula and training courses specific to each EMS level. After proper training on ECG interpretation, follow-up studies across levels of EMS can be carried out to determine the time needed for a refresher ECG training course”.

Note:

We have also made some minor corrections throughout the manuscript to accommodate the above amendments. 

 

References

1. Alrumayh AA, Mubarak AM, Almazrua AA, Alharthi MZ, Alatef DF, Albacker TB, et al. Paramedic ability in interpreting electrocardiogram with ST-segment elevation myocardial infarction (STEMI) in Saudi Arabia. J Multidiscip Healthc. 2022:1657-65. PMID: 35959233.

2. Alanazi A, Wark S, Fraser J, Nagle A. Utilization of prehospital emergency medical services in Saudi Arabia: An urban versus rural comparison. J Emerg Med Trauma Acute Care. 2020;9:1-7. http://dx.doi.org/10.5339/jemtac.2020.9.

3. Alanazi A, Wark S, Fraser J, Nagle A. Emergency medical services in rural and urban Saudi Arabia: A qualitative study of Red Crescent emergency personnel’s perceptions of workforce and patient factors impacting effective delivery. Health Soc Care Community. 2022;30:e4556-e64. https://doi.org/10.1111/hsc.13859.

4. AlShammari T, Jennings P, Williams B. Evolution of emergency medical services in Saudi Arabia. J Emerg Med Trauma Acute Care. 2017;4:1-11. https://doi.org/10.5339/jemtac.2017.4.

5. Saudi Red Crescent Authority. National scope of practice (Version 1.2). SRCA 2022:1-53 (Available from https://www.srca.org.sa/en/statistics/emergency-medical-services-blog/). 

6. Cajavilca C, Varon J. Willem Einthoven: The development of the human electrocardiogram. Resuscitation. 2008;76:325-8. PMID: 18164799. 

7. Kashou A, May A, DeSimone C, Noseworthy P. The essential skill of ECG interpretation: How do we define and improve competency? Postgrad Med J. 2020;96:125-7. PMID: 31874907.

8. Delewi R, Van de Hoef TP, Hirsch A, Robbers LF, Nijveldt R, van der Laan AM, et al. Pathological Q waves in myocardial infarction in patients treated by primary PCI. JACC Cardiovasc Imaging. 2013;6:324-31. PMID: 23433932. 

9. Beygui F, Castren M, Brunetti ND, Rosell-Ortiz F, Christ M, Zeymer U, et al. Pre-hospital management of patients with chest pain and/or dyspnoea of cardiac origin. A position paper of the Acute Cardiovascular Care Association (ACCA) of the ESC. Eur Heart J Acute Cardiovasc Care. 2020;9:59-81. PMID: 26315695.

---

## [Editor Report · Decision Letter 1]

2 Oct 2023

Electrocardiographic interpretation by emergency medical services professionals in Saudi Arabia: A cross sectional study

PONE-D-23-21375R1

Dear Dr. Alalwan,

We’re pleased to inform you that your manuscript has been judged scientifically suitable for publication and will be formally accepted for publication once it meets all outstanding technical requirements.

Kind regards,

Ibrahim Marai, MD

Academic Editor

PLOS ONE

Additional Editor Comments (optional):

Thank you for you for you reply and modifications
---

## [Editor Report · Acceptance letter]

9 Oct 2023

PONE-D-23-21375R1 

Electrocardiographic interpretation by emergency medical services professionals in Saudi Arabia: A cross sectional study 

Dear Dr. Alalwan:

I'm pleased to inform you that your manuscript has been deemed suitable for publication in PLOS ONE. Congratulations! Your manuscript is now with our production department. 

Kind regards, 

on behalf of

Dr. Ibrahim Marai 

Academic Editor

PLOS ONE